# Deep Learning for Fingerprint-Based Outdoor Positioning via LTE Networks

**DOI:** 10.3390/s19235180

**Published:** 2019-11-26

**Authors:** Da Li, Yingke Lei

**Affiliations:** School of Electronic Countermeasures, National University of Defense Technology, Hefei 230000, China; ryxar@163.com

**Keywords:** outdoor positioning, fingerprint positioning, deep learning, Resnet, transfer learning

## Abstract

Fingerprint-based positioning techniques are a hot research topic because of their satisfactory accuracy in complex environments. In this study, we adopted the deep-learning-based long-time-evolution (LTE) signal fingerprint positioning method for outdoor environment positioning. Inspired by state-of-the-art image classification methods, a novel hybrid location gray-scale image utilizing LTE signal fingerprints is proposed in this paper. In order to deal with signal fluctuations, several data enhancement methods are adopted. A hierarchical architecture is put forward during the deep neural network (DNN) training. First, the proposed positioning technique is pre-trained by a modified Deep Residual Network (Resnet) coarse localizer which is capable of learning reliable features from a set of unstable LTE signals. Then, to alleviate the tremendous collection workload, as well as further improve the positioning accuracy, by using a multilayer perceptron (MLP), a transfer learning-based fine localizer is introduced for fine-tuning the coarse localizer. The experimental data was collected from realistic scenes to meet the requirement of actual environments. The experimental results show that the proposed system leads to a considerable positioning accuracy in a variety of outdoor environments.

## 1. Introduction

In recent years, smartphone-based positioning has been attracting attention due to the increasing number of equipped sensors and rapid development of various positioning techniques. The successful application of satellite navigation positioning technology, such as the global positioning system (GPS), has enabled people to travel all over the world freely. Although GPS may be the primary choice of outdoor positioning, it does not have a satisfactory performance in complex environments, such as crowded cities or those with unfavorable weather. When GPS signals experience none-line-of-sight (NLOS) propagation, these signals are blocked by buildings or trees, and the GPS-based positioning service can greatly deteriorate. In addition, GPS exhibits a huge energy consumption.

The broad-scale popularity of Long-time-evolution (LTE) signals and multiple sensors equipped on user equipment (UE), such as smartphones, has opened up a new boulevard for outdoor positioning. Low power sensors built in UE have enabled us to conveniently collect positioning information. Additionally, unique signal characteristics of the environment collected from chips and corresponding locations represent an alternative way of locating a UE. The unique characteristics of features as signatures matched against pre-defined geotagged signatures are regarded as fingerprint-based positioning [1].

Compared to other positioning technologies utilizing LTE signals or satellite signals, such as the angle of arrival (AOA), time of arrival (TOA), time difference of arrival (TDOA), GPS, and hybrid positioning methods [2,3], smartphone fingerprint-based positioning has a variety of merits. First, low-power consumption chips built in smartphones draw a much lower energy, even when smartphones constantly receive signals. Second, most smartphone-based positioning requires no additional hardware or infrastructure, which can greatly save costs. In addition, fingerprint-based positioning technology can acquire favorable positioning effects, even when signal NLOS propagation exists [4].

Many complex clues are hidden in our surroundings, so the purpose of fingerprint-based positioning is to effectively find the geo-tag hidden clues and then leverage them to determine the location of UE [5]. Owing to the constantly increasing number of sensors in UE, three kinds of fingerprint-based positioning types exist in the literature: visual fingerprint, motion fingerprint, and signal fingerprint [1]. With the rapid development of image processing techniques, various sorts of image features can be extracted as the geo-tag features. Additionally, retrieval techniques, such as Google Goggles [6], have been proposed to search the considerable image dataset utilizing visual features, and the geo-tag features can then be matched against the location [7]. However, the visual-based positioning method has a great defect because of the tremendous collection workload in the offline phase and huge search volume in the positioning phase, so it is difficult to achieve large-scale outdoor positioning. Due to the increasing number of motion sensors, such as accelerometers and electronic compasses, UE can leverage them to perform motion recognition. The basic idea of motion fingerprint-based positioning is to build a fingerprint dataset which combines the motion sensor’s features with the geographic location and then matches the sensor’s features against the location in the positioning phase. Due to the limited positioning information, motion fingerprint is usually used as the assist location information [8,9]. The tremendous number of smartphones with built-in chips and popularization of LTE networks have opened up a new avenue for location services. Furthermore, the signal fingerprint-based positioning techniques have shown a great localization accuracy, even when signals are blocked by trees and buildings. The basic idea of the signal fingerprint-based technique is to find the geo-tag signal features, such as channel state information (CSI), received signal strength indication (RSSI), reference signal receiving power (RSRP), and reference signal receiving quality (RSRQ), and then match them against a pre-defined signal database to find the location of UE [10,11,12]. Due to the small range of variations, RSRQ is usually used as the assist positioning information. Researchers have tried to leverage CSI between LTE base stations (BSs) and UE to achieve a high accuracy positioning, but this method needs very expensive instruments and has tremendous workloads. Therefore, these adverse aspects make it difficult to apply for large-scale outdoor positioning applications.

In recent years, deep learning has made great progress in many fields and achieved state-of-the-art performances in feature extraction, image recognition, and so on. Therefore, in this paper, Resnet and transfer learning are investigated to provide a low surveying cost and satisfactory positioning accuracy in signal NLOS propagation environments. Owing to the state-of-the-art performance of deep learning in image classification, a novel fingerprint image using RSSI, RSRP, and RSRQ to represent the features of locations is well-constructed. The proposed method combines the RSSI, RSRP, and RSRQ in a single image. The three signals can be collected at the same time during the sampling process. In addition, owing to the merit of the orientation-free property, as well as the rapid sampling rate, the difficulty of collecting the three signals is eradicated.

The main problem in applying deep learning to outdoor positioning is the instability of LTE signals. Therefore, in this paper, the designed network adopts pre-training and fine-tuning of the two-level hierarchical architecture to improve the positioning accuracy and save the workload of collection. After the fingerprint image dataset is well-constructed, in the training stage, the Resnet model is first formed through pre-training the dataset. After obtaining the best positioning accuracy, we freeze the parameters of the Resnet model. Then, by using the prior Resnet model information, transfer learning is utilized to fine-tune the positioning accuracy. In order to fully extract the signal features from the instability LTE signal training dataset, several data enhancement methods are adopted to ameliorate the method. First, the size of each picture is expended into 224 × 224, which allows Resnet to extract features better. Second, in order to increase the diversity of pictures, some of the images are enlarged by 1.25 times, and another way is to randomly rotate the image by 15°. Besides, in the batch normalization phase, a momentum item is added to reduce the vibration time and accelerate the convergence of the Resnet. Then, multi-layer perception is further attached to Resnet for the purpose of increasing Resnet’s learning ability. For the matching stage, a probabilistic method is proposed to predict the location of UE.

The main contributions of this paper can be summarized as follows. Compared to other fingerprint-based positioning methods, our system first leverages a hybrid location gray-scale fingerprint image for positioning. We propose a positioning system based on Resnet and transfer learning two-level hierarchical architecture for outdoor positioning. The proposed system can overcome the LTE signal fluctuations and offer satisfactory positioning accuracy. Considering the numerous classification points, several data enhancement methods are adopted, and we leverage enough training and preserve the best testing accuracy epoch model method to prevent the overfitting problem and maximize the positioning accuracy. The system is tested in a real environment to verify the preliminary previous theory. The experiments convincingly show that the proposed positioning system reaches a satisfactory performance in a variety of outdoor environments.

The rest of this paper is arranged as follows: Section 2 reviews the LTE signal-based positioning techniques; in Section 3, the proposed positioning system is overviewed; Section 4, Section 5 and Section 6 describe the Fingerprint-image construction, DNN training module, DNN positioning module; Section 7 describes the experimental scenarios and steps of experimental implementation; and finally, Section 8 describes conclusions and future work.

## 2. Related Works

The broad demand for positioning services has spurred the development of positioning techniques. Several papers have proposed solutions for estimating the UE location using LTE networks.

Range-based methods leveraging TOA, TDOA, AOA, or hybrid methods, such as TOA/AOA, have been proposed to offer positioning services [2,3]. However, these methods have many limitations as they need additional expensive hardware and the positioning accuracy heavily depends on the environment, as well as synchronization with the base station. Compared to ranged-based positioning methods, fingerprint-based positioning technology shows a higher accuracy, even in the presence of signal NLOS propagation. A variety of fingerprint-based positioning methods exist in the literature. 

K-Nearest-Neighbor (KNN) and Weighted K-Nearest-Neighbor (WKNN) have been utilized to match data from the constructed database for positioning [13,14]. Owing to the shallow model learning ability, it has an unsatisfactory positioning accuracy. Ye et al. [10] proposed a neural network assist positioning method to improve the positioning effect. In order to further improve the positioning accuracy, a CSI-based method has been proposed [12,15]. However, this method requires dedicated hardware, specific CSI signals, and a burdensome workload, so it is inconvenient for massive outdoor applications. There is also literature like [16,17,18] that has attempted to leverage shallow neural networks for UE positioning. Owing to the limited learning ability and fluctuation of LTE signals, these methods do not achieve a satisfactory accuracy. In order to solve this problem, Ma et al. [18] proposed an LTE signal fluctuation elimination method to further improve the positioning accuracy. In recent years, with the development of computers’ calculation ability, some solutions have proposed deep learning architectures for positioning. Because of the satisfactory learning ability, these methods can achieve a great accuracy [4,12,19,20]. However, the proposed methods have only been tested in indoor environments or signal LOS propagation outdoor environments, and the collection process requires a burdensome workload. Hence, whether large-scale outdoor applications are feasible is still a problem. 

Different from the aforementioned positioning techniques, our proposed technique requires no additional expensive hardware or complex analysis signals. The ubiquitous LTE signals are utilized, by combining RSSI, RSRP, and RSRQ in a common fingerprint image. The proposed technique is free of orientation information. Therefore, it has no requirements for UE attitudes. In addition, transfer learning is utilized for cutting off the tremendous workload. Therefore, our system is more practical and cost-effective than other positioning methods.

## 3. Proposed System Architecture 

As Figure 1 shows, the proposed fingerprint-based positioning system consists of the following modules: LTE signal pre-processing, fingerprint classification, fingerprint-image construction, DNN training, and DNN positioning. Besides, the positioning information containing LTE signals is featured with its geo-tag label. After the process of DNN training, the DNN model parameter database stores the Resnet and transfer learning model for different positioning sites. The sensor LTE data leveraged in our positioning system include RSSI, RSRP, and RSRQ.

### 3.1. LTE Signal Pre-Processing

The raw signal value of the base station’s RSSI and RSRP is rectified between −40 and −140 dBm. The purpose of pre-processing is to modify the signal value and make it adaptable to an RGB image based on $v_{i}=\left(RSSI+150\right)$, $v_{p}=\left(RSRP+150\right)$. In each grid, the RSRQ value is modified as follows: (1)$$v_{q}=\sqrt{\frac{1}{N}\sum_{i=1}^{N}\left(RSRQ-\bar{RSRQ}\right)},$$
where $\bar{RSRQ}=\frac{1}{N}\sum_{i=1}^{N}RSRQ_{i}$. i indicates the number of signals, N is the total number of signals collected in each grid.

### 3.2. Fingerprint Classification

In order to construct the LTE signal map match against the area of interest, UE collect surrounding signals walking along sampling lines. The purpose of fingerprint classification is to divide the positioning area into multiple grids of the same size. After the work of collecting signals is completed, the fingerprint classification modules equally divide the area of interest into dozens of hundred grids. When positioning with the probabilistic method, the size of the grid determines the positioning accuracy [21,22]. Hence, in order to achieve a satisfactory positioning accuracy, the divided grids should not be too large.

### 3.3. Fingerprint Image Construction

Owing to the different data lengths in each fingerprint grid, the task of the fingerprint-image construction module is to modify all the fingerprint data so that they are the same length. This module is used both in the training phase and positioning phase. 

### 3.4. DNN Training

Due to the instability of LTE signals, environment signal features need to be fully learned. Therefore, several data enhancement methods are proposed and a hierarchical architecture of DNN training is adopted by the proposed system.

The DNN-based localizer has two steps, where the Resnet is first pre-trained on the fingerprint-image of the training database. After getting the best positioning effect of Resnet, the Resnet coarse localizer model is kept for transfer learning. For the fine-tuning step, by leveraging the prior knowledge of the trained Resnet, the proposed system adds another multi-layer perception (MLP) for transfer learning.

### 3.5. DNN Positioning

The online positioning phase consists of the Resnet coarse localizer and transfer learning-based fine localizer.

Coarse positioning: Resnet is first utilized to automatically learn the constructed image-feature. The output of the Resnet is the probabilities of the right grid locations.

Fine positioning: Transfer learning is taken into account to further learn the image features in order to achieve a better positioning accuracy.

## 4. Fingerprint-Image Construction

Different from traditional methods of extracting positioning signal features, this positioning system proposed a novel image-based method leveraging computer vision to extract signal features. Generally, an ordinary RGB image contains three dimensional matrixes, which are red, green and blue, respectively. If the values for each channel are the same, this image is termed as grey scale image. In this positioning system, the collected sensor data series are conveyed into grey scale image. 

After collecting a series of sensor data, we leveraged three column vectors consisting the RSSI, RSRP, and RSRQ information to form an image. In order to use image as the input of DNN, we normalized the image dimensions to the same size. And in each grid, we divide the data set into several sub-data sets and build fingerprint image based on the sub-data sets. Thus, for the R channel, part of the fingerprint image can be represented as follows:(2)$$\varphi ={\left[v_{i1},v_{i2},\ldots ,v_{i10}\right]}^{T}$$
(3)$$\psi ={\left[v_{p1},v_{p2},\ldots ,v_{p10}\right]}^{T}$$
(4)$$\gamma ={\left[v_{q1},v_{q2},\ldots ,v_{q10}\right]}^{T}$$
(5)$$v_{q1}=v_{q2}=\ldots =v_{q10}=v_{q}$$
(6)F=[φ,ψ,γ]
where 10 is the length of sub-data dimension. F is the matrix representation of the R channel. After the construction of R channel, G and B channels are constructed in the same way as R. Therefore, a grey scale image can be well constructed by using the three channels matrix.

## 5. DNN Training Module

The positioning performance may vary greatly due to the different training methods. Therefore, in this paper, we present enough training and preserving the best testing accuracy epoch model method to maximize the positioning accuracy. The DNN module, in this paper, consists of a Resnet part and transfer learning part, and we trained these two parts separately. 

The number of training epochs has a great influence on the DNN performance. Few training epochs will make it hard for DNN to fully learn the data set feature, leading to inaccurate positioning. In comparison, enough training will cause an overfitting problem [23]. Therefore, in order to solve this problem, first, we disrupted the order of all fingerprint images. Then, we used the first 80% of the data as the training set and the remaining 20% as the test set. For each training epoch, the accuracy of the test set was evaluated and the best training epoch was reserved. Figure 2 shows that as the training epoch increased, the DNN module continuously learnt data set features. The test set accuracy increased first, when the overfitting problem occurred, and then, the test set accuracy began to drop. Therefore, in this system, we first fully trained the DNN model, and then chose the best positioning accuracy module as the final model.

In addition, several data enhancement methods are adopted in this paper. First, the picture was standardized to 224 × 224, which allowed Resnet to better learn image features. Second, some of the images were enlarged by 1.25 times, whilst another way is to randomly rotate the original image by 15°. This method can enhance the richness and diversity of the dataset. Batch normalization was further added to each substructure of Resnet. Besides, in the batch normalization item, a momentum item was added to reduce the vibration time and accelerate the convergence of Resnet. By leveraging these techniques, the proposed system can further improve the positioning accuracy. Transfer learning has many merits. First, it can save training costs to a large extent. Second, the prediction results for small data sets can be significantly improved. The general definition of transfer learning is: storing knowledge gained while solving one problem and applying it to a different but related problem. In this paper, the idea of transfer learning is reflected in freezing the parameters of pre-trained model and then using the pre-trained model’s prior knowledge to train the customized model [23]. Specifically, after Resnet is trained, the parameters of the trained Resnet are frozen. And then a fully connected layer is added to the Resnet for training. During this training process, only the parameters of the MLP are changed. Therefore, the storing knowledge is obtained from the pre-trained Resnet, and we applied it for the training of MLP. And the purpose of this method is to fully study the signal features. After the training of Resnet, transfer learning added a fully connected layer for fine-tuning the positioning result. In order to maximize the positioning accuracy, enough training and preserving of the best testing accuracy epoch model method was also adopted.

## 6. DNN Positioning Module

The proposed coarse localizer is a modified Resnet-based probabilistic estimator consisting of eighteen residual modules and two fully-connected layers. Benefitting from transfer learning, the fine localizer used the prior Resnet knowledge and an MLP with several hidden layers to further improve the positioning accuracy. In this section, the DNN algorithm and positioning module are introduced.

### 6.1. Deep Residual Network Introduction

Current neural network knowledge shows that the deeper the level of the network, the better the learning ability. However, sometimes, due to gradient dispersion and gradient explosion issues, the deep neural network has a higher test error compared to the shallow neural network. This phenomenon is called degradation [24]. In order to solve this problem, He, et al. [25] proposed the residual learning concept. Suppose that a submodule of a neural network needs to learn the target mapping of H(x), while H(x) may be too complex to learn. Therefore, instead of directly learning the target map, let the submodule learn the residual F(x)=H(x)−x. Therefore, the original target map changes into F(x)+x, and the submodule is composed of two parts: direct linear mapping x→x and nonlinear mapping F(x). If the direct mapping of x→x is optimal, the neural network will set the weight parameters of the nonlinear map F(x) to 0. The deep residual neural network is mainly composed of multiple residual learning modules. The residual learning module is shown in Figure 3.

In this paper, as is shown in Figure 4, the coarse localizer is a modified Resnet consisting of 18 residual units, an average pooling layer, a flatten layer, and an MLP with two hidden layers. Each unit consists of a convolutional layer, batch normalization layer, and ReLu layer, respectably. When learning fingerprint image features, the convolution layer slides through the entire image. In each step, the convolution window calculates the dot product between the kernel vector and the image vector inside the kernel. Therefore, the output of the convolutional layer is
(7)$$\{\begin{array}{l} x_{j}^{la}=f\left(u_{j}^{la}\right)\\ u_{j}^{la}=\sum_{i\in M_{j}}x_{i}^{la-1}*k_{ij}^{la}+b_{j}^{la} \end{array}$$
where $u_{j}^{la}$ is the net activation of the jth channel of the convolutional layer l. It is obtained by convolution summation of the previous layer output feature map $x_{j}^{la}$ and adding the offset term. $x_{j}^{la}$ is the output of the jth channel of the convolutional layer l. f(⋅) is the activation function, and in this paper, the ReLu function is used as the activation function. $M_{j}$ represents the input feature map subset used to calculate $u_{j}^{la}$. $k_{ij}^{la}$ is the convolution kernel matrix, $b_{j}^{la}$ is the offset to the convolutional feature map, and “*”represents the convolution symbol.

The ReLu nonlinear activation function can be represented as
(8)$$x_{j}^{lb}=f\left(u_{j}^{la}\right)=max(0,u_{j}^{la}).$$

In order to accelerate the training speed and prevent gradient dispersion, batch normalization is adopted. This process is as follows:(9)$$\{\begin{array}{l} x_{j}^{lc}=f\left(u_{j}^{lb}\right)\\ u_{j}^{lb}=\frac{x_{j}^{lb}-E\left(x_{j}^{lb}\right)}{\sqrt{Var\left(x_{j}^{lb}\right)}} \end{array},$$
where $E\left(x_{j}^{lb}\right)$ and $Var\left(x_{j}^{lb}\right)$ are the mean value and variance of $x_{j}^{lb}$, respectively. This will output the characteristic map $x_{j}^{lb}$ of the previous layer in a normal distribution.

After propagation of the 18 residual units, average pooling is utilized to down sample the previous feature map, which can reduce the number of parameters and prevent the overfitting problem. The average filter is leveraged to the sub-area of the last layer feature map and the average feature is taken as the new output feature map. This process can be presented as
(10)$$\{\begin{array}{l} x_{m}^{k}=f\left(u_{m}^{k}\right)\\ u_{m}^{k}=\beta_{m}^{k}down\left(x_{m}^{k-1}\right)+b_{m}^{k} \end{array},$$
where $u_{m}^{k}$ is the net activation of the mth channel of the down sampling layer k. It is obtained by down sampling the weight of the previous layer output characteristic map $x_{m}^{k-1}$. β is the weighting factor of the down sampling layer, $b_{m}^{k}$ is the bias term of the down sampling layer, and down(⋅) represents the down sampling function. It divides the input feature map $x_{m}^{k-1}$ into multiple non-overlapping n×n image blocks by sliding down the sampling window, and then calculates the mean value of the pixels in each image block. Therefore, the output image is reduced by n times in both dimensions [26].

The purpose of the flatten layer is to convert the multidimensional input into a one-dimensional input, and realize the transition from the average pooling layer to the fully connected layer.

Then, an MLP with two hidden layers is leveraged to further extract features. The output of the fully connected layer l can be obtained by weighting the input and passing the activation function: (11)$$\{\begin{array}{l} x^{l}=f\left(u^{l}\right)\\ u^{l}=w^{l}x^{l-1}+b^{l} \end{array},$$
where $u^{l}$ is the net activation of the fully connected layer l. $w^{l}$ and $b^{l}$ are the weighting factor and bias term of the fully connected network, respectively.

In the back-propagation phase, this system minimizes cross entropy loss between the prediction label and the true label provided by the last fully connected layer. This minimization process is served by an adaptive moment estimation (Adam) algorithm for the purpose of adjusting the value of weights. 

Finally, Softmax regression and argmax are utilized to evaluate the extracted high-level features and give the result of probabilistic position estimates:(12)$$P(M=m_{i}|v)=\frac{exp(-w_{i}x-b_{i})}{\sum_{i}exp(-w_{i}x-b_{i})},$$
where $m_{i}$ represents the ith grid position, $w_{i}$ denotes the weights between the output layer and the previous layer, and $b_{i}$ is the bias of the output layer. Then, argmax is used to extract the highest probability grid position as the final position. When calculating the total positioning accuracy, first, the positioning model calculates the estimated position of each test sample, and then compares it with the real position. Then the positioning accuracy can be obtained by the following function:(13)$$P=\frac{n}{N}$$
where N is the total number of fingerprint images in the test set, and n is the number of fingerprint images that are correctly estimated.

### 6.2. Transfer Learning Introduction

After Resnet is well-trained, transfer learning is leveraged to fine-tune the neural network. Transfer learning can be divided into two steps. First, the parameters of the trained Resnet are frozen. Second, a fully connected layer is added to the Resnet. Figure 5 shows the overall structure of the proposed DNN.

When training the transfer learning, the parameter of Resnet remains unchanged. Another MLP consisting of several hidden layers is leveraged to further improve the positioning accuracy. Its feed-forward and backward propagation are the same as the aforementioned MLP layer. In addition, cross entropy loss and the Adam algorithm are utilized to modify the neural network. The Resnet and MLP use the same training set and test set for the purpose of comparing the coarse localizer and the fine localizer positioning accuracy.

## 7. Experiments and Results

As Figure 6 shows, measurement campaigns were conducted in an outdoor environment divided into dozens of grids, and the red line indicates our test site, which consists of pedestrians, cars, buildings, and trees. It is worth mentioning that this environment consists of a variety of scenes, which poses a huge challenge to the positioning ability of the proposed system. Instead of leveraging any expensive hardware, like Universal Software-defined Radio Peripheral (USUP) device [10], a smartphone was leveraged to collect signals. One person walked around the grids and held a smartphone with an Android system, which was equipped with a ubiquitous built-in chip that could receive the LTE signal. By using the Cellular-Z application, we could receive the RSSI, RSRP, and RSRQ from one LTE base station at a sample rate of 100–140 per minute. We divided the grid into 20 × 20 m. The way of collecting the dataset also had an impact on positioning, and in order to study the impact of mobility on its positioning accuracy, we took a smartphone and moved around to collect the signals in each grid [27]. The goal of the test process was to find out the current location of UE through the proposed system and compare it with the truth location. 

The learning rate is an important hyperparameter in deep learning, which determines whether the objective function can converge to a local minimum and when it converges to a minimum. As is shown in Table 1, as the learning rate increases, the test accuracy increases first and then decreases. During the training process, we found that when the learning rate is set too large, the gradient oscillates greatly around the minimum. When the learning rate is set too small, the convergence process becomes very slow, which greatly increases the training time and the accuracy is not very satisfactory. This is probably because a low learning rate will cause the neural network to fall into the local minimum solution. As can be observed from Table 1, we chose the learning rate of 1 × 10^−3^ as the hyperparameter of the neural network.

The grid size determines the accuracy of the positioning. If the grid size is too small, the positioning accuracy will decrease rapidly. Additionally, if the grid size is too large, it will lose the meaning of positioning [4]. As is shown in Table 2, we tested the impact of the grid size on the positioning accuracy. As the grid size becomes larger, the test accuracy continues to increase. When the grid size exceeds 20 × 20 m, the accuracy improvement is not obvious. Therefore, 20 × 20 m was chosen as the grid size. We also compared the impact of different algorithms on the test accuracy in Table 2. In addition, Figure 7 and Figure 8 show the positioning accuracy of shallow models and deep models, respectively. And the proposed DNN greatly outperforms KNN, WKNN, SVM, and GRNN. This is because these shallow models have a limited learning ability and the proposed DNN has an excellent modeling ability. Furthermore, it is more capable of learning reliable features and thus more robust when estimating locations with these features. In Figure 8, we can see the proposed DNN outperforms other deep models. This shows that the algorithm proposed in this paper has stronger learning ability.

The number of hidden layers and hidden units has a great influence on the performance of the neural network. Therefore, we compared the effects of different hidden layers and hidden units of transfer learning on the positioning accuracy. μ donates the number of hidden layers. Figure 9 indicates that as the number of hidden units increases, a greater positioning accuracy can be obtained. However, when the MLP goes deeper, the positioning performance becomes worse, which is probably because the deeper neural network causes an overfitting problem and makes the gradients hard to propagate. As is shown in Figure 9, the best positioning accuracy is obtained with three hidden layers and 200 units per hidden layer. Therefore, an MLP, which has three hidden layers, is used to build the transfer learning in this system.

When the Adam algorithm is used to optimize the network, a smaller batch size will cause the training error to oscillate greatly, and the neural network is difficult to converge. In comparison, a larger batch size can lead to the neural network generalization performance deteriorating. Besides, the proportion of the training set to the total data set will affect the training accuracy. As shown in Figure 10 and Figure 11, we tested the effects of transfer learning, the batch size, and the size of the training sample on the training accuracy. γ indicates the proportion of the training set to the total data set. η denotes the batch size. As is shown from these two pictures, in most cases, transfer learning can improve the positioning accuracy greatly, and the best positioning accuracy is 93.6%. It is not surprising that the larger the training set, the better the positioning performance. The results suggest that γ=0.8 works well. It can be observed that as the batch size increases, a better positioning accuracy can be obtained. However, an excessive batch size has an adverse effect on the positioning performance. Hence, γ =0.8, η = 32 is chosen to build the localizer in this system. In addition, considering the way of collecting signals and the satisfactory positioning accuracy, the proposed DNN localizer can positioning user equipment when users are moving.

In order to test the effect of transfer learning on the positioning accuracy of small samples, we collected a small number of signal samples in the same area and tested the positioning accuracy of the proposed coarse localizer and fine localizer. As can be seen from Figure 12, transfer learning can improve the positioning accuracy under the condition of small amount of training data. Figure 10 and Figure 11 show that when the training data is sufficient, transfer learning can still improve the positioning accuracy. Therefore, transfer learning can further improve positioning accuracy based on the coarse localizer.

## 8. Conclusions

The basic idea of LTE signal positioning techniques is to find the location of a smartphone device by comparing its signal feature received from LTE BSs with a pre-defined geotagged database of signal features. In this paper, we have introduced a deep learning system for LTE signal-based outdoor positioning, which has rarely been included in previous literature. Additionally, a novel fingerprint representation method which converts LTE signals into fingerprint images was developed. In order to overcome LTE signal fluctuation, several data enhancement methods were adopted, and we leveraged two-level hierarchical architecture deep learning to learn reliable features from the unstable training data sets, which makes it feasible for large-scale outdoor applications. Resnet is first utilized to learn the fingerprint image features, and then, by using Resnet’s prior knowledge, transfer learning is adopted to further improve the outdoor positioning accuracy. The experimental results revealed that the proposed system can offer a satisfactory positioning accuracy in a variety of outdoor environments. Information technology (IT) has an important impact on a company’s agility and long-term development. IT infrastructure capability, IT business spanning capability, and IT proactive stance play an important role in a firm’s agility [28]. Additionally, positioning techniques could be a good option for the development of IT capability, especially for traveling companies. This would greatly improve travelers’ online information satisfaction, which could bring considerable profits to the company and builds a competitive advantage in uncertain environments [29,30]. For future work, we intend to work with local companies to achieve engineering goals, as well as generate business benefits and social benefits for the company.

## Figures and Tables

**Figure 1 sensors-19-05180-f001:**
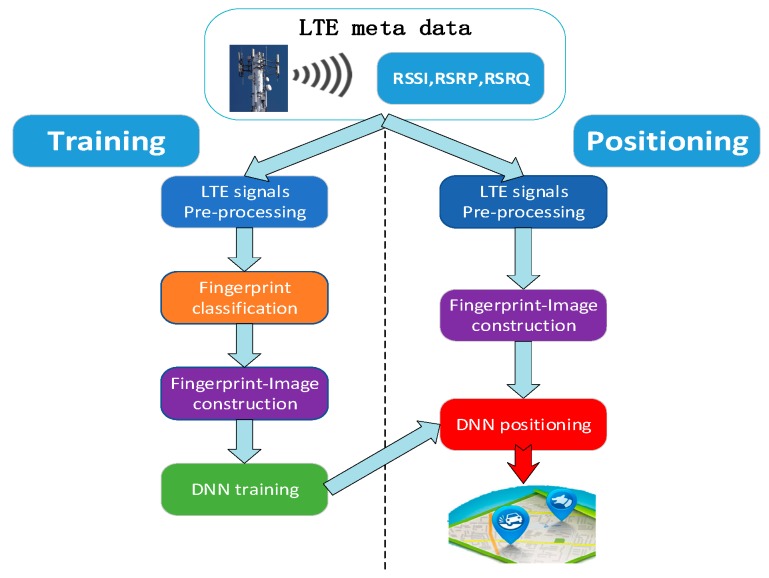
The overall architecture of our proposed outdoor positioning system based on deep neural network (DNN) and fingerprint-image learning. DNN stands for Resnet coarse localizer and multi-layer perception (MLP)-based transfer learning fine localizer.

**Figure 2 sensors-19-05180-f002:**
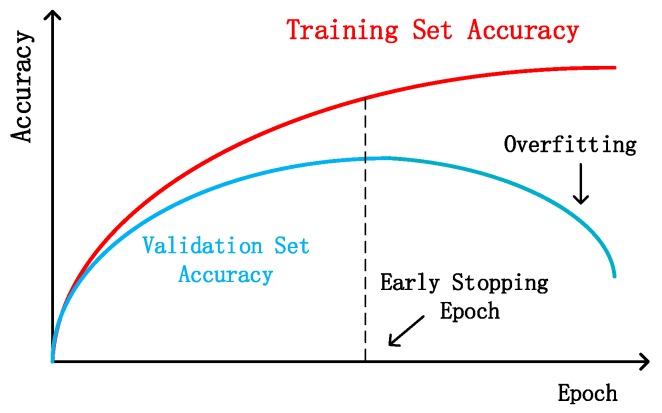
Schematic diagram of deep neural network (DNN) training set accuracy and test set accuracy change with the training epoch.

**Figure 3 sensors-19-05180-f003:**
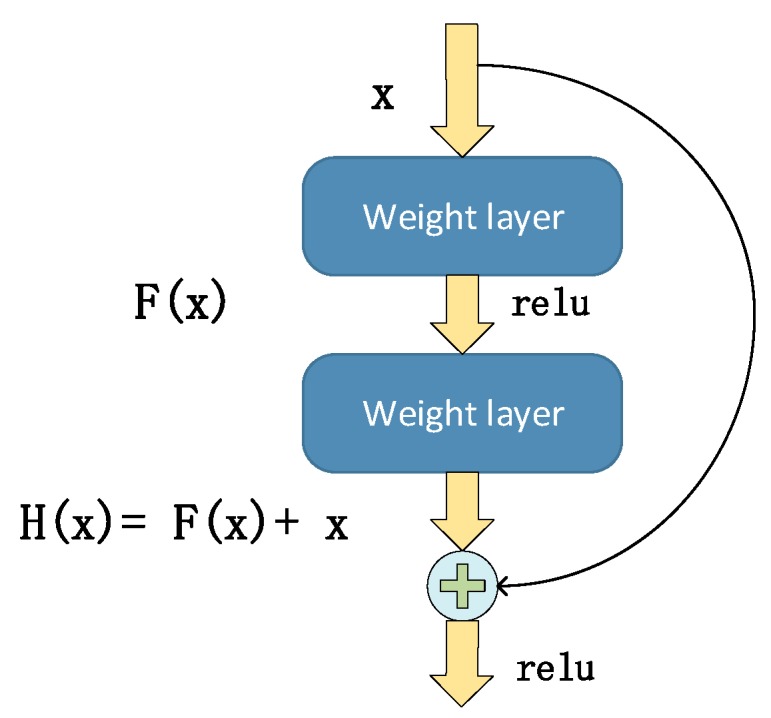
Schematic diagram of the substructure unit of Resnet.

**Figure 4 sensors-19-05180-f004:**
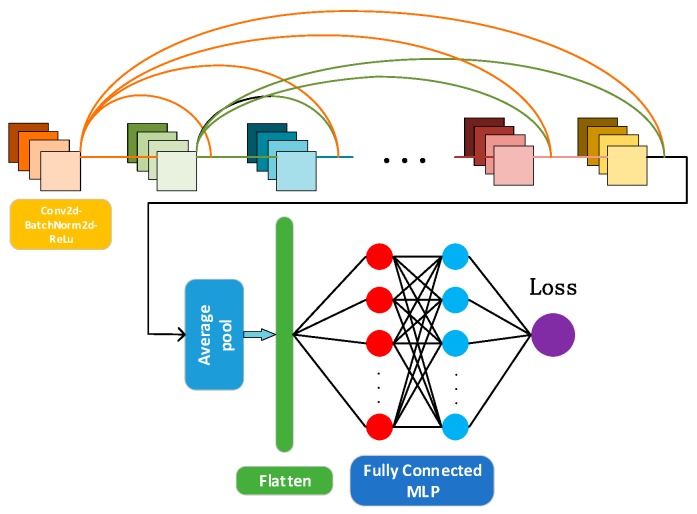
The structure of the proposed Resnet coarse localizer.

**Figure 5 sensors-19-05180-f005:**
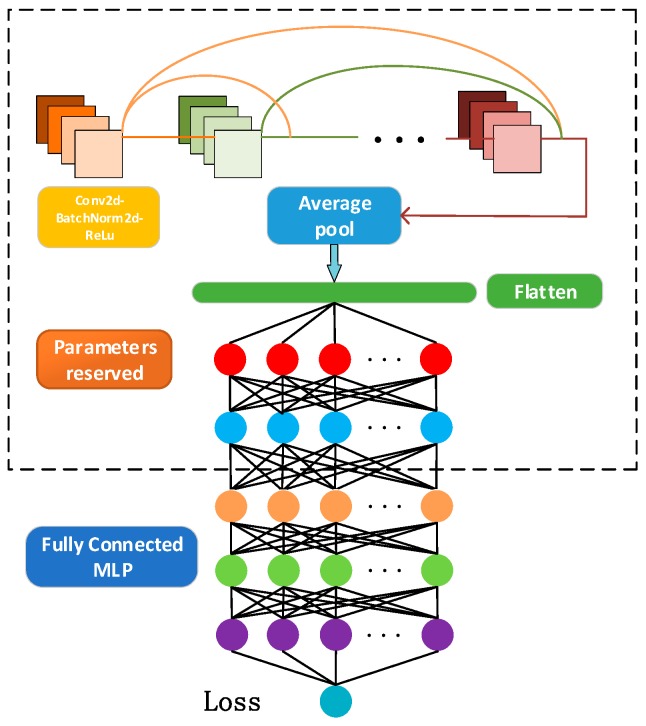
The proposed transfer learning fine localizer structure.

**Figure 6 sensors-19-05180-f006:**
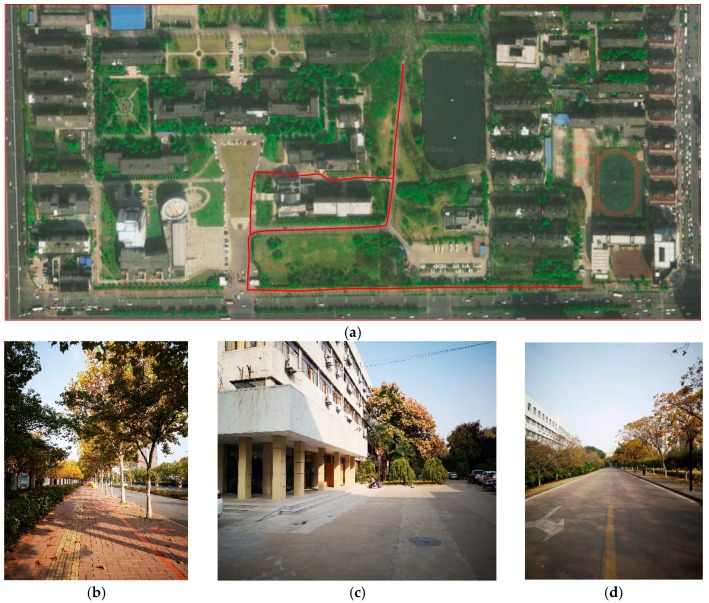
The satellite map and photographs of the outdoor positioning area. (**a**) Satellite map of the positioning area. (**b**–**d**) Real scene of the positioning area.

**Figure 7 sensors-19-05180-f007:**
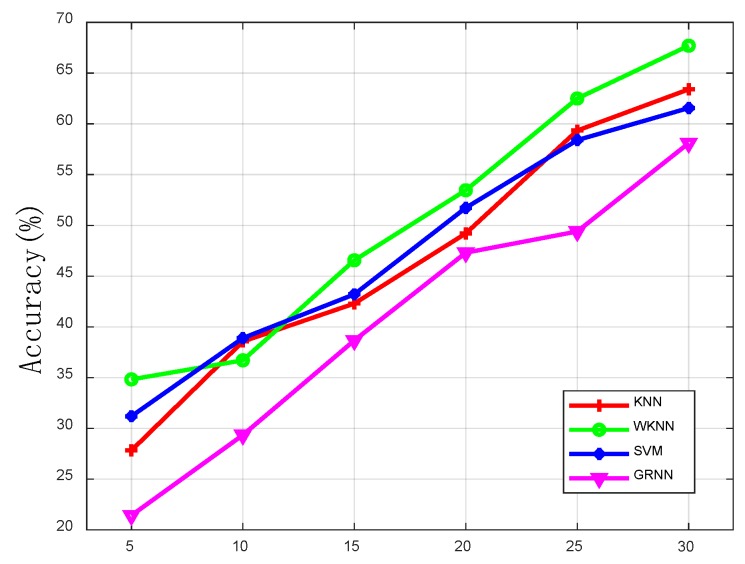
The outdoor positioning accuracy with shallow models.

**Figure 8 sensors-19-05180-f008:**
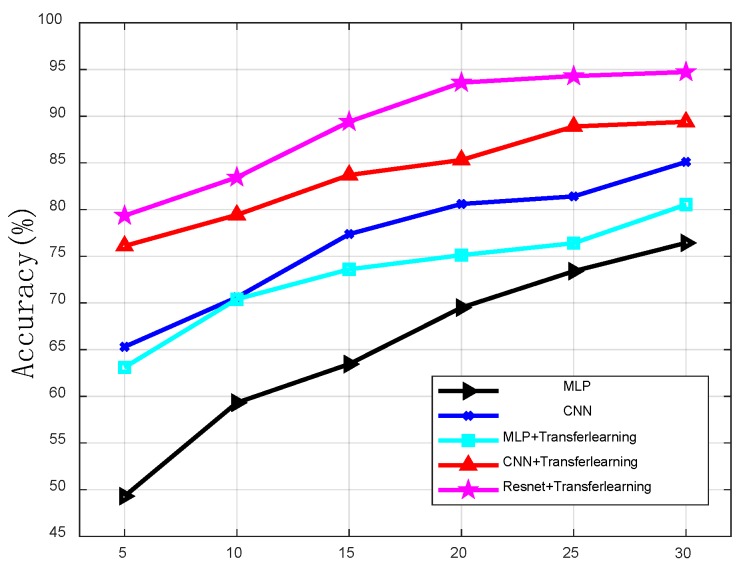
The outdoor positioning accuracy with deep models.

**Figure 9 sensors-19-05180-f009:**
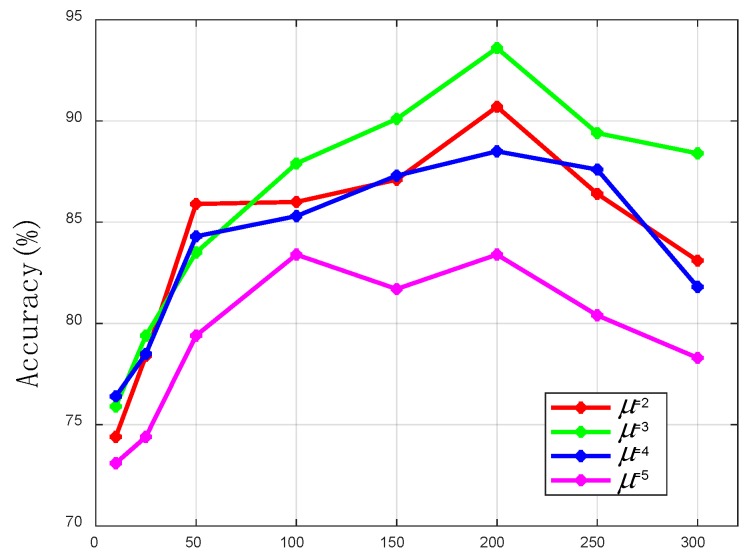
The outdoor positioning accuracy of fine localizer with respect to μ and the number of hidden units. μ donates the number of hidden layers.

**Figure 10 sensors-19-05180-f010:**
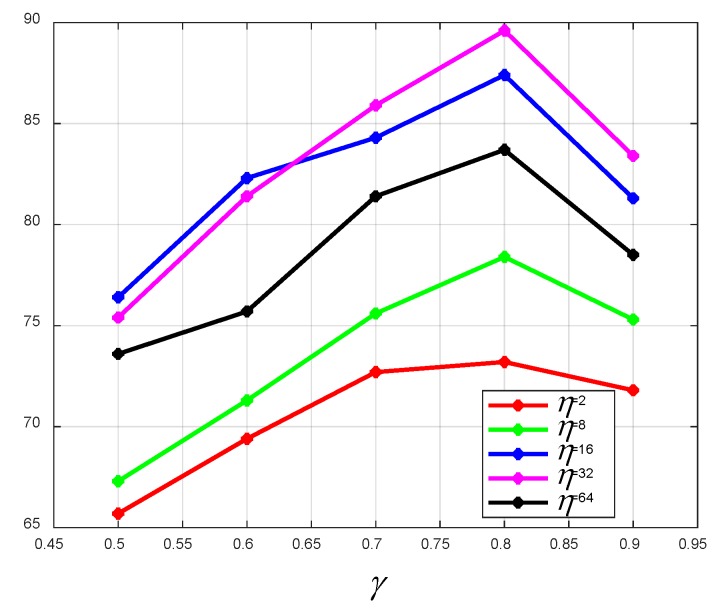
The outdoor positioning accuracy of the coarse localizer with respect to γ and η. γ indicates the proportion of the training set to the total data set. η denotes the batch size.

**Figure 11 sensors-19-05180-f011:**
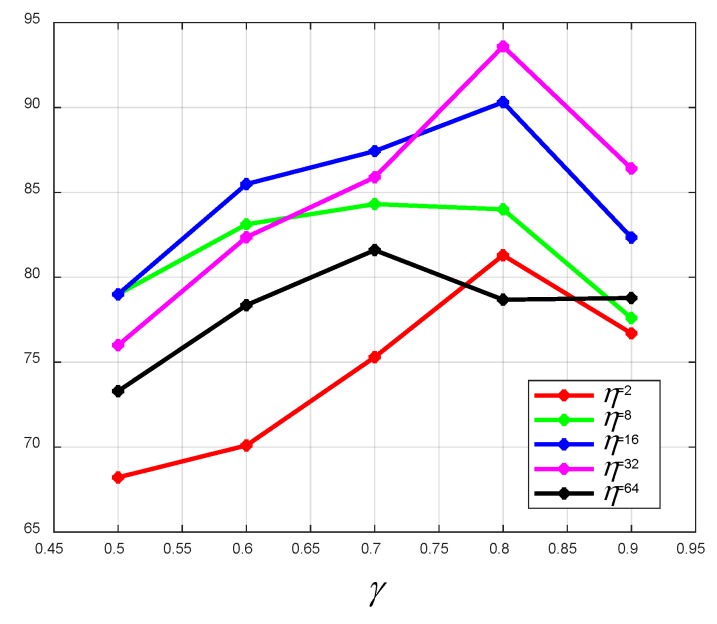
The outdoor positioning accuracy of the fine localizer with respect to γ and η. γ indicates the proportion of the training set to the total data set. η denotes the batch size.

**Figure 12 sensors-19-05180-f012:**
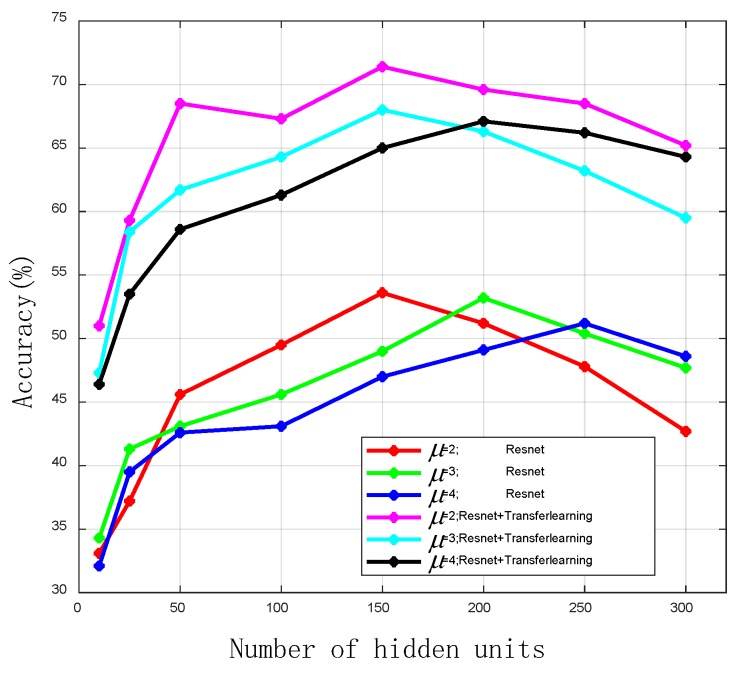
The outdoor positioning accuracy of coarse localizer and fine localizer with small amount of data. μ donates the number of hidden layers.

**Table 1 sensors-19-05180-t001:** Test accuracy (%) with different learning rates.

Learning Rate	Resnet Accuracy (%)	Transfer Learning Accuracy (%)
1 × 10^−1^	66.7	45.68
1 × 10^−2^	75.6	69.78
1 × 10^−3^	86.3	93.6
1 × 10^−4^	74.6	85.3
1 × 10^−5^	76.56	79.35

**Table 2 sensors-19-05180-t002:** Test accuracy (%) of different algorithms with different grid sizes.

Grid Size (m)	5 × 5	10 × 10	15 × 15	20 × 20	25 × 25	30 × 30
KNN (%)	27.83	38.56	42.3	49.21	59.34	63.4
WKNN (%)	34.83	36.71	46.56	53.46	62.5	67.7
SVM (%)	31.2	38.9	43.21	51.73	58.4	61.56
GRNN (%)	21.43	29.35	38.67	47.31	49.4	58.1
MLP (%)	49.3	59.31	63.45	69.5	73.4	76.43
CNN (%)	65.3	70.6	77.37	80.6	81.41	85.1
MLP + Transferlearning (%)	63.1	70.4	73.6	75.12	76.4	80.53
CNN + Transferlearning (%)	76.1	79.43	83.7	85.31	88.9	89.4
Resnet + Transferlearning (%)	79.35	83.45	89.41	93.6	94.3	94.73

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
