# Peer review of "Deep Learning for Fingerprint-Based Outdoor Positioning via LTE Networks"

_sensors, 2019, doi:10.3390/s19235180_

Round 1
Reviewer 1 Report
The paper proposes a fingerprint based outdoors positioning technique based on hybrid location gray scale image utilizing LTE signal fingerprints and deep learning. The data used in the study are too few, as also noted by the authors, therefore, the reviewer is skeptical about the reliability of the results. IT would be a good idea to first of all attempt collect more data and test their idea on a variety of environments and also, to provide a comparison in the experiments section with other state-of-the-art approaches (this part is a big weakness of the study). As claimed in the introduction, the proposed technique is better to similar studies as it does not require specialized expensive hardware and it is less costly, However, it would be nice to see a proof of such a claim in some way.
Lastly, although the paper is not difficult to read, it is weak in presentation as it requires a good review of the english use.
Reviewer 2 Report
Overall the paper is very interesting and I enjoyed reading it. The method is well executed and the structure and presentation are good. One aspect though that I believe is very week is the discussion of results. There is practically no discussion in the paper regarding the theoretical and practical implications of your work. It would be very interesting for readers if you could talk about the practical relevance of your work particularly in combination to other scientific work. Perhaps you could discuss about the relevance for IT-enabled agility in operations as talked from a business angle from Lu and Ramamurthy (2011) and Mikalef and Pateli (2017). This would show how your work has relevance to a higher level also.
Lu, Y., & K.(Ram) Ramamurthy. (2011). Understanding the link between information technology capability and organizational agility: An empirical examination. MIS quarterly, 931-954.
Mikalef, P., & Pateli, A. (2017). Information technology-enabled dynamic capabilities and their indirect effect on competitive performance: Findings from PLS-SEM and fsQCA. Journal of Business Research, 70, 1-16.
Reviewer 3 Report
The paper tackles the problem of localization of a mobile user in a outdoor complex environment. This problem is challenging since existing GPS based solutions do not work well when network signals are blocked by obstacles and GPS based devices consume huge amount of energy. The authors propose to use LTE signals to compute a location fingerprint since it does not have the problem faced by GPS. However, LTE are not quite stable. Authors propose to deal with this problem with a hierarchical approach. First a modified Resnet is trained as a coarse localizer and then a transfer learning based fine localizer is introduced. The authors show the performance of their proposed DNN. The experiments seems to be sound. However, I can highlight a few details:
1) In general, the article is very well written and it is easy to read. just a few minor grammar issues were found.
2) Although, some experimental results are given and it shows the effectiveness of the proposed solution. The authors do not provide results from other approaches (baseline) to effectively compare their proposed solution. To me, this is the main drawback of the article.
3) It is not clear how the training data is divided for the Resnet and the MLP.
4) Although, a grid 20x20 mts seems to be fine but is there any reason to choose this grid size.
5) The authors do not show the impact of mobility in the accuracy of the proposed localizer
Reviewer 4 Report
This paper proposes an outdoor positioning scheme based on ' Resnet+ MLP'. It first exploits Resnet for coarse positioning, and then MLP is used for further fine-tuning the results. However, the following problems or questions exist:
There are many grammatical problems in this paper, and English expression needs to be improved. Accuracy seems to be the only evaluation indicator in this paper, but the author does not present its definition and calculation method. In Sections 3.2 and 3.3, the introduction about ' Fingerprint Construction' and 'Fingerprint image Construction' are too brief to understand the input format of Resnet, more figures and detailed descriptions are needed. This article mentioned for several times that the use of transfer learning-based fine localizer can reduce the tremendous collection workload, but no experiment results is presented to support it. The general definition of transfer learning is: storing knowledge gained while solving one problem and applying it to a different but related problem. Does the scheme proposed in this article have this property? whether the model trained in a certain area can be used for positioning in other areas? The formula (1) in Section 3.1 is unclear(what’s the meaning of i and N?). There are many different variables share the same symbol, which makes readers confuse though they have been defined several times. Some subscripts/superscripts will be helpful. The pictures and mathematical expressions in this paper are too blurry. There is a mistake for Chapter label (missing Chapter 4).
Round 2
Reviewer 3 Report
After reviewing again the arcticle, I think the authors have addressed all my comments. There are still some minors details, different fonts, that can be addressed again by the authors. A few minor details:
In page 3, line 16 says "Section4, 5 and 6 detailed"
Page 7, line 16 "accounting for 80% and 20%, respectively. When dividing
the training set and test set, first, we disrupted the order of all fingerprint images. Additionally, we used the first 80% of the data as the training set and the remaining 20% as the test set" ( the description of the traininig and testing sets is repeated)
A big empty space at page 4. Title 3.2 should be moved to the next page.
Author Response
Thank you very much for your insightful comments. We have made corrections to these contents. And please see the attachment.
